# Sentinel-2 Application to the Surface Characterization of Small Water Bodies in Wetlands

**Jesús Pena-Regueiro [1], Maria-Teresa Sebastiá-Frasquet [1,\*], Javier Estornell [2] and Jesús Antonio Aguilar-Maldonado [1,3]**

[1] Research Institute for Integrated Management of Coastal Areas, Universitat Politècnica de València, C/ Paraninfo, 1, 46730 Grau de Gandia, Spain; jepere@doctor.upv.es

[2] Geo-Environmental Cartography and Remote Sensing Group, Universitat Politècnica de València, Camí de Vera s/n, 46022 Valencia, Spain; jaescre@upv.es

[3] Facultad de Ciencias Marinas, Universidad Autónoma de Baja California, Ensenada 22860, Mexico; jeagmal@upv.es

\* Correspondence: mtsebastia@hma.upv.es

**Abstract:** Developing indicators to monitor environmental change in wetlands with the aid of Earth Observation Systems can help to obtain spatial data that is not feasible with in situ measures (e.g., flooding patterns). In this study, we aim to test Sentinel-2A/B images suitability for detecting small water bodies in wetlands characterized by high diversity of temporal and spatial flooding patterns using previously published indices. For this purpose, we used medium spatial resolution Sentinel-2A/B images of four representative coastal wetlands in the Valencia Region (East Spain, Mediterranean Sea), and on three different dates. To validate the results, 60 points (30 in water areas and 30 in land areas) were distributed randomly within a 20 m buffer around the border of each digitized water polygon for each date and wetland (600 in total). These polygons were mapped using as a base map orthophotos of high spatial resolution. In our study, the best performing index was the NDWI. Overall accuracy and Kappa index results were optimal for −0.30 threshold in all the studied wetlands and dates. The consistency in the results is key to provide a methodology to characterize water bodies in wetlands as generalizable as possible. Most studies developed in wetlands have focused on calculating global gain or loss of wetland area. However, inside of wetlands which hold protection figures, the main threat is not necessarily land use change, but rather water management strategies. Applying Sentinel-2A/B images to calculate the NDWI index and monitor flooded area changes will be key to analyse the consequence of these management actions.

**Keywords:** remote sensing; wetlands; NDWI; Kappa index; overall accuracy

## 1. Introduction

Wetlands, and especially coastal wetlands, are listed amongst the most threatened ecosystems suffering from anthropogenic activities [1]. These ecosystems provide a wide range of ecosystem services. Among others, they are an important freshwater reserve and a source for groundwater recharge, and in coastal areas they provide defence against marine intrusion. Wetland hydrologic regime (e.g., flooded area and flooding duration) has a direct effect on nutrient dynamics at a watershed scale, but it also impacts greenhouse gas emissions and carbon cycles in the wetlands themselves [2]. The crucial role of flooding extent to wetland functioning and carbon storage has been underlined by several studies [3]. The spatial and temporal variation in flooded areas can be high, and it is due to hydrological processes (e.g., precipitation, evapotranspiration), but also to human activities [3]. Impacts on hydrological processes can affect other ecosystems functions such as

groundwater recharge and nutrient cycling [4] or species distributions and composition [5]. So, monitoring spatial-temporal dynamics of flooded areas is both important for water management and biodiversity conservation [6].

In recent years, monitoring programs relied on in situ detectors to collect data used by regulatory agencies and research institutions. However, gauge measurements offer little information about spatial patterns like flooding status [6,7]. On the other side, remotely sensed data can provide spatial maps with different accuracy depending on the sensor [7]. Remote sensing has already proved to be a useful tool to acquire spatial and temporal information about wetlands [8,9] and it has the potential to provide the information needed for accurate wetland inventory, assessment, and monitoring [5]. Detection and analysis of wetlands using satellite images are based mainly on supervised and unsupervised classification and the definition of water indices and their subsequent classification using thresholds [10,11]. For supervised classification, several algorithms can be found such as random forest [12], support vector machines [13], and artificial neural networks [14]. The development of Earth Observation (EO) satellites of high spatial resolution and the emergence of Unmanned Aerial Vehicle (UAV) contributed to the definition of new approaches to map wetlands at sub-meter scale such as object-oriented algorithms [15]. The high resolution of images registered by UAVs allow to monitor and to extract the flooding surface with detail and to develop ecological indicators [16].

For the approach based on water indices, a simple global threshold can be applied to classify water pixels using atmospherically corrected satellite images of different data and places [17]. Images from satellite sensor of low spatial resolution such as AVHRR (Advanced Very High Resolution Radiometer) and MODIS (Moderate Resolution Imaging Spectroradiometer) have been used to monitor flood extent by differentiating flooded/non-flooded or mixed pixels [6,7]. Studies using these sensors focused mainly on relatively large wetlands covering at least 50 km$^2$, and to a lesser extent to smaller wetlands (<25 km$^2$) [7]. Other sensors with higher spatial resolution, such as those on-board Landsat MSS/TM/ETM+ and SPOT, have been used for smaller wetlands monitoring [3]. The coarser spatial resolution sensors have the advantage of a higher temporal resolution and more frequent observations than higher spatial resolution sensors. According to Huang et al. [7] in arid, semi-arid, and Mediterranean environments, about 30% of Ramsar-listed seasonal wetlands are small-sized, with a minimum size of 10 ha, which can show a patchy distribution of water bodies. From 2015 onward, Sentinel-2A/B images are available (ESA), with high temporal resolution and bands of 10 m that allow to explore the suitability of these images to extract small-sized water bodies that cannot be mapped using Landsat and MODIS images.

In recent years, there is a growing interest in developing indicators to monitor environmental change in wetlands through remote sensing [18]. Some of the original constraints (e.g., insufficient spatial and temporal resolution) of this technique have been overcome with the last satellites launching. However, we still have the constraint of developing indicators that can be global, and non-specific for a type of wetland or location.

The aim of this study is to test Sentinel-2A/B images suitability for detecting small water bodies in wetlands characterized by high diversity of temporal and spatial flooding patterns using previously published indices.

## 2. Materials and Methods

### 2.1. Study Area

Mediterranean wetlands are ecosystems identified for priority protection by the European Union (EU) [19]. These ecosystems have been studied as prototypes of coastal wetlands where urban and agricultural pressure compete directly with environmental water uses [20]. The Mediterranean is one of the regions with the highest pressure on wetlands, and especially in coastal areas [21,22]. We selected 4 representative coastal wetlands in the Valencia Region (East Spain, Mediterranean Sea), from North to South: Prat Cabanes-Torreblanca, Sagunto, Safor, and Pego-Oliva (Figure 1). These wetlands are included in both the Valencian Wetlands Inventory, and in the Spanish Wetlands

Inventory (included in 2011 by Resolution of 9 March Dirección General de Medio Natural y Política Forestal (Spanish Official Gazette (BOE), Number 71, 24 March 2011)). Pego-Oliva and Prat Cabanes-Torreblanca are also declared Natural Parks. The importance of these wetlands is also recognized at international level, all of them are Special Protection Areas (SPAs) for birds and Sites of Community Importance (SCIs) (Habitats Directive, European Council Directive 92/43/EEC). In addition, Prat Cabanes-Torreblanca and Pego-Oliva are Ramsar Sites (Ramsar Convention).

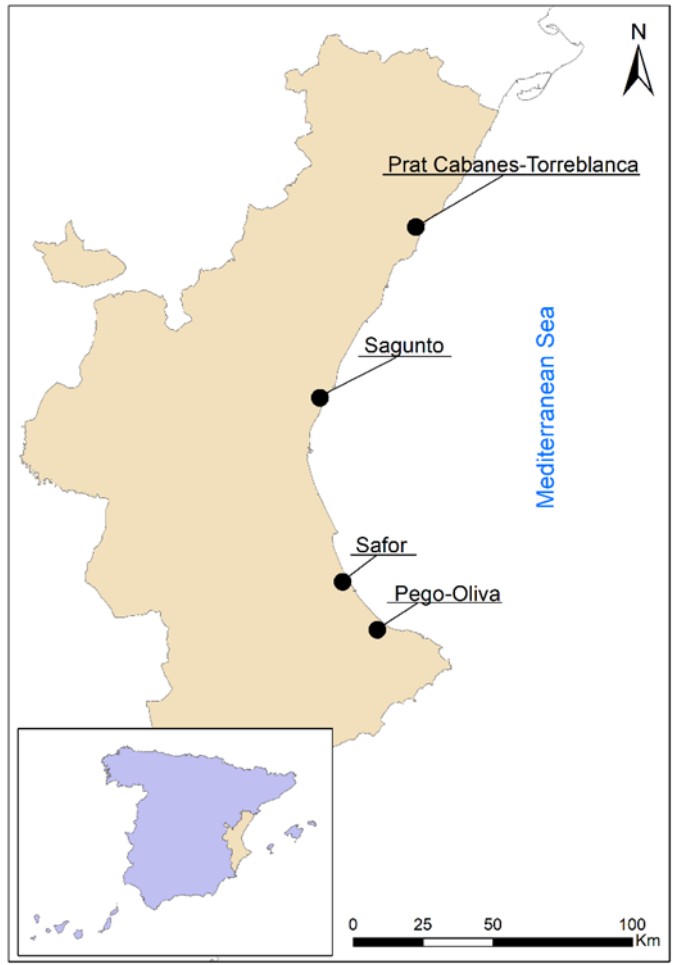

**Figure 1.** Wetlands location in the Valencian Region (East Spain).

These wetlands are set on detrital aquifers formed from the Quaternary fluvial sedimentation that filled the coastal plains and separated from the sea with beach barriers. Detrital aquifers are, in turn, fed by Mesozoic karstic aquifers in the near limestone reliefs. The wetlands are fed by groundwater discharges and depend on them to keep permanent surface of water [23]. Phreatic level is subject to seasonal variations but usually emerges very close to the surface. Groundwater upwelling creates water ponds, known as "ullals", in the wetland environment. The wetland discharge is mainly due to natural drainage to the sea through rivers and to groundwater pumping for several purposes. Specifically, the shallow phreatic level in these unconfined aquifers causes problems of root asphyxia to citrus crops and flooding of urban areas. To prevent this problem, freshwater from the aquifer is pumped into the sea through irrigation channels [24]. The cycle of the wetlands and the size of flooded areas is highly dependent on precipitation, which rapidly infiltrates in the karstic aquifers and discharges to the detrital aquifers [23,25]. However, anthropogenic management is also very important. The precipitation regime is characterized by a very marked seasonality, the highest rainfall in autumn, and secondarily in spring; during the summer there is a strong drought [26]. The average annual precipitation shows a spatial gradient, Pego-Oliva

(southernmost area) annual average is 841.3 mm, while Prat Cabanes is 447 mm (northernmost area). Table 1 shows annual precipitation data from the closest meteorological station [27].

**Table 1.** Annual precipitation data (mm) and location of meteorological stations. Available in: http://riegos.ivia.es/datos-meteorologicos.

| Wetland | Year | | | Coordinates | |
|---|---|---|---|---|---|
| | 2016 | 2017 | 2018 | UTM X | UTM Y |
| Prat Cabanes | 231.09 | 414.74 | 200.54 | 768076.000 | 4447370.000 |
| Sagunto | 226.60 | 662.61 | 234.38 | 732200.000 | 4392210.000 |
| Safor | 285.29 | 791.36 | 405.74 | 738207.000 | 4316410.000 |
| Pego-Oliva | 380.57 | 924.31 | 360.97 | 767731.000 | 4298290.000 |

Water depth in the flooded areas of these wetland range from wet soil to 70 cm [28], only the "ullals" have higher depths. Table 2 shows the diversity of habitats present in these wetlands and included in the EU Habitats Directive (Council Directive 92/43/EEC) according to the European Nature Information System [29].

**Table 2.** Natura 2000 Habitat types present in the study areas according to the European Union (EU) Habitats Directive Annex I classification (source: https://eunis.eea.europa.eu/index.jsp). Empty cell means the habitat is not present in that wetland.

| Code | Annex I Habitat types | Prat Cabanes | Sagunto | Safor | Pego-Oliva |
|---|---|---|---|---|---|
| | | Cover (ha) | | | |
| 1150 | Coastal lagoons | 19.40 | 79.74 | | 12.55 |
| 1410 | Mediterranean salt meadows (*Juncetalia maritimi*) | 97.00 | 46.49 | | 150.60 |
| 1420 | Mediterranean and thermo-Atlantic halophilous scrubs (*Sarcocornetea fruticosi*) | 19.40 | 27.25 | | |
| 1510 | Mediterranean salt steppes (*Limonietalia*) | | 4.82 | | |
| 3150 | Natural eutrophic lakes with Magnopotamion or Hydrocharition -type vegetation | | | 248.97 | 25.10 |
| 3160 | Natural dystrophic lakes and ponds | | | 186.73 | 12.55 |
| 3170 | Mediterranean temporary ponds | 19.40 | | | |
| 3280 | Constantly flowing Mediterranean rivers with Paspalo-Agrostidion species and hanging curtains of *Salix* and *Populus alba* | | | 62.24 | 25.10 |
| 5330 | Thermo-Mediterranean and pre-desert scrub | | 0.35 | | 125.5 0 |
| 6110 | Rupicolous calcareous or basophilic grasslands of the Alysso-Sedion albi | | | | 25.10 |
| 6220 | Pseudo-steppe with grasses and annuals of the Thero-Brachypodietea | | | | 62.75 |
| 6420 | Mediterranean tall humid grasslands of the Molinio-Holoschoenion | 194.00 | 0.37 | 62.24 | 150.60 |
| 6430 | Hydrophilous tall herb fringe communities of plains and of the montane to alpine levels | | | 62.24 | 12.55 |
| 7210 | Calcareous fens with *Cladium mariscus* and species of the *Caricion davallianae* | 388.00 | 133.60 | 622.43 | |

*2.2. Image Processing*

Sentinel-2A/B images processed at level 1C were obtained from Copernicus (https://scihub.copernicus.eu/dhus/#/home) and EarthExplorer (https://earthexplorer.usgs.gov). The atmospheric correction was done with Sen2Cor tool (version 02.05.05) using SNAP software (ESA,

version 6.0.0). The following reference parameters were defined: aerosol and MID LAT (Auto); ozone (value 0, determined automatically by the processor); Bidirectional Reflectance Distribution Function (BRDF) correction (value 21, standard recommended value); BRDF lower (value 0.22, standard value of the lower limit of the correction factor BRDF); visibility (40 km, appropriate value for the Iberian Peninsula); altitude (2 m above sea level). Images of high spatial resolution were used for validation (Table 3) [11,30]. These images were obtained from the Valencian Cartography Institute (ICV) Orthophoto 2017 and 2018 CC BY 4.0 © Institut Cartogràfic Valencià, Generalitat (spatial resolution 0.25 m, http://www.icv.gva.es/va/) and Google Earth ©. The dates of these images were the closest to Sentinel-2A/B image acquisitions.

**Table 3.** List of images used in the study by date.

| Wetland | Orthophoto | | Sentinel-2A/B |
|---|---|---|---|
| | **Data** | **Spatial Resolution** | **Data** |
| Prat Cabanes-Torreblanca | 28 July 2018 | 0.25 m | 30 July 2018 |
| | 5 July 2017 | 0.25 m | 5 July 2017 |
| Sagunto | 8 July 2018 | 0.25 m | 5 July 2018 |
| | 18 June 2017 | 0.25 m | 15 June 2017 |
| | 17 November 2016 | n. i.* | 17 November 2016 |
| Safor | 13 June 2018 | 0.25 m | 20 June 2018 |
| | 18 August 2017 | 0.25 m | 4 August 2017 |
| | 11 November 2016 | n. i.* | 7 November 2016 |
| Pego-Oliva | 13 June 2018 | 0.25 m | 20 June 2018 |
| | 11 November 2016 | n. i.* | 7 November 2016 |

* n. i. no information available for Google Earth images.

The official cartography of these protected areas (Valencian Wetland Inventory) was used to delimitate each wetland (Figure 2). The methodology was applied to the polygons classified as natural areas in the SIOSE (Information System on Land Use in Spain) cartography. For each date and area, we delimited the water and non-water polygons. Water polygons smaller than 100 m$^2$ were excluded, considering the maximum spatial resolution of Sentinel-2A/B bands used in this study. The water and non-water polygons were delineated through visual examination using as a base map high-resolution image (orthophoto) and was done with the software ArcGis 10.5 (ESRI 2016. ArcGIS Desktop: Release 10.5 Redlands, CA: Environmental Systems Research Institute). In Figure 3 delineated polygons can be observed (blue colour). The visual delimitation was possible thanks to the high spatial resolution of the orthophotos (0.25 m).

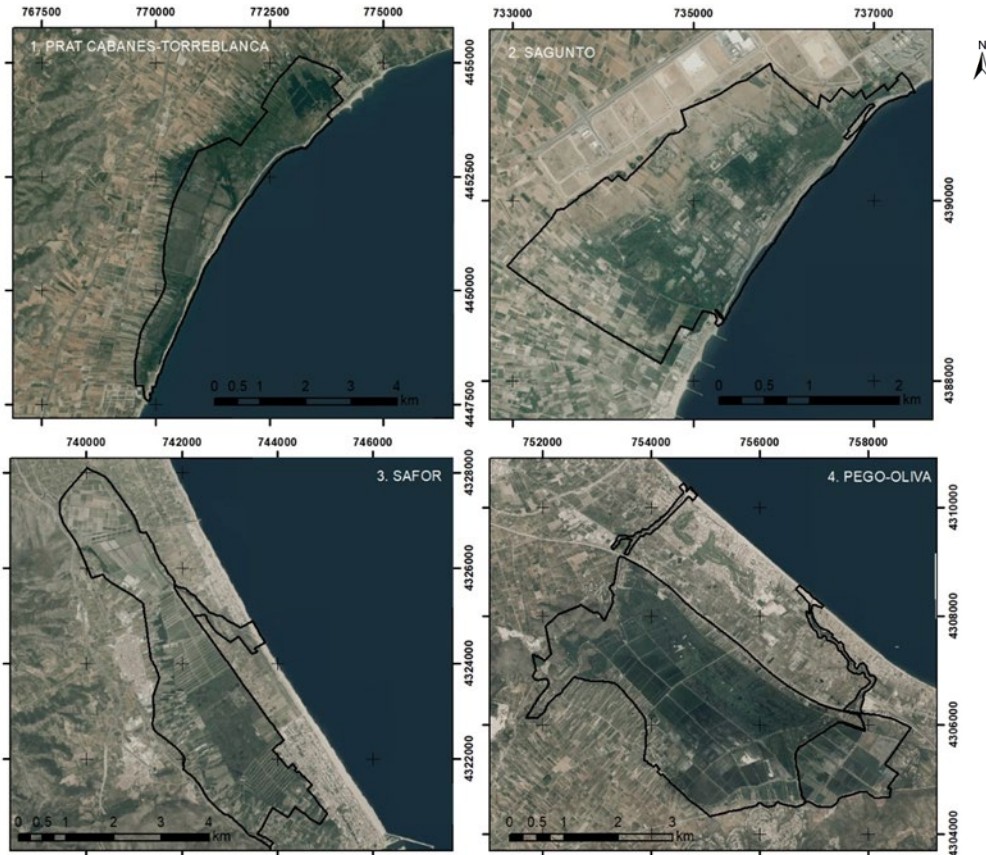

**Figure 2.** Wetlands area according to the official delimitation of the protected area (Valencian Wetland Inventory). Spatial Reference System: European Terrestrial Reference System 1989, UTM coordinates Zone 30S.

The spectral information was extracted from the Sentinel-2A/B images to calculate the seven spectral indices shown in Table 4 for each wetland polygon (Figure 2). The choice of indices was based on literature review. These indices classify as water/non-water according to a threshold value, but different authors propose different thresholds for the same indices. We aimed to define a unique threshold that can be as global as possible with optimum results. So, for each date and area we tested all the thresholds from −0.50 to 0.50 with a 0.05 step, except for the AWEI(NSH) (Automated Water Extraction Index, No Shadow) and the AWEI(SH) (Automated Water Extraction Index, shadow) indices whose thresholds ranged from −50 to −5000, and the step is detailed in the results section.

**Table 4.** Calculated spectral indices.

| INDEX | EQUATION | SOURCE | SENTINEL-2 BANDS |
|---|---|---|---|
| NDWI | [(GREEN − NIR) / (GREEN + NIR)] | [31] | [(B03 − B08) / (B03 + B08)] |
| MNDWI | [(GREEN − SWIR) / (GREEN + SWIR)] | [32] | [(B03 − B11) / (B03 + B11)] |
| CEDEX | (NIR / RED) − (NIR / SWIR) | [33] | (B05 / B04) − (B05 / B11) |
| RE-NDWI | [(GREEN − NIR) / (GREEN + NIR)] | [34] | [(B03 − B05) / (B03 + B05)] |
| AWEI(SH) | BLUE + 2.5× GREEN − 1.5 × (NIR + SWIR) − 0.25− SWIR | [17] | [B02 + 2.5 × B03 − 1.5 × (B08 + B011) − 0.25 × B12] |
| AWEI (NSH) | 4 × (GREEN−MIR) − (0.25 × NIR + 2.75 × SWIR) | [17] | [4 × (B03 − B11) − (0.25 × B08 + 2.75 × B12)] |
| B_BLUE | (BLUE − NIR) / (BLUE + NIR) | This study | (B02 − B08) / (B02 + B08) |

To validate the results obtained from the Sentinel-2A/B images, we designed a random sampling of 60 points for each date and wetland. The ground control points were distributed randomly within a 20 m buffer around the border of each digitized water polygon. These features were mapped using as a base map orthophotos of high spatial resolution. A large number of both water points nearshore (30 points for each wetland and date, 300 in total) and surrounding non-water land points (30 points for each wetland and date, 300 in total) were selected in these areas with high spectral variability, which makes a total of 600 points for validation. We selected the number of points according to the general guideline provided by Congalton and Green [35], who recommended a minimum of 50 samples for each map class for maps of less than 1 million acres in size and fewer than 12 classes. For all these points, we compared the classification of each index (7 indices in Table 4) and each threshold, with the ground-truth images, to assess correct classifications. Overall accuracy and Kappa index were calculated for each random sampling. Overall accuracy was obtained by dividing the number of pixels correctly classified by the total number of pixels sampled [35]. Kappa index was calculated according to Congalton's [36] equation. The best index and threshold were selected according to overall accuracy and Kappa index results.

## 3. Results

Overall accuracy and Kappa index results are represented in Figures 3, 4, 6, and 7 with a colour scale. Shaded in yellow appear the indices and thresholds with poorest performance, that is when the classification system from the Sentinel-2A/B images fails to meet the reality defined from the ground-truth images. Shaded in red appear the indices and thresholds with best performance (closer to 1), that is when the classification system from the Sentinel-2A/B images matches the reality defined from the ground-truth images.

In Figure 3, the overall accuracy results are presented for all seven tested indices. The tested thresholds ranged are detailed in the methodology section (step detailed in Figure 3). The best overall accuracy result (0.89) is for NDWI index with −0.30 threshold. The other indices showed lower overall accuracy results for all the tested thresholds (≤0.85).

| Thresholds | Index | | | | | | Thresholds | Index | |
|---|---|---|---|---|---|---|---|---|---|
| | NDWI | MNDWI | CEDEX | RE−NDWI | B_Blue | | | Awei (sh) | Awei (nsh) |
| 0.50 | 0.54 | 0.52 | 0.62 | 0.50 | 0.54 | | | | |
| 0.45 | 0.54 | 0.53 | 0.62 | 0.50 | 0.54 | | | | |
| 0.40 | 0.54 | 0.53 | 0.62 | 0.50 | 0.54 | | −50 | 0.62 | 0.53 |
| 0.35 | 0.55 | 0.53 | 0.62 | 0.50 | 0.54 | | −100 | 0.63 | 0.53 |
| 0.30 | 0.55 | 0.53 | 0.62 | 0.50 | 0.54 | | −200 | 0.65 | 0.53 |
| 0.25 | 0.56 | 0.53 | 0.62 | 0.50 | 0.55 | | −300 | 0.66 | 0.53 |
| 0.20 | 0.57 | 0.53 | 0.62 | 0.50 | 0.55 | | −400 | 0.67 | 0.53 |
| 0.15 | 0.59 | 0.55 | 0.62 | 0.51 | 0.56 | | −500 | 0.68 | 0.54 |
| 0.10 | 0.60 | 0.57 | 0.62 | 0.53 | 0.56 | | −600 | 0.68 | 0.55 |
| 0.05 | 0.63 | 0.58 | 0.62 | 0.54 | 0.57 | | −700 | 0.69 | 0.56 |
| 0.00 | 0.66 | 0.60 | 0.62 | 0.56 | 0.60 | | −800 | 0.70 | 0.57 |
| −0.05 | 0.69 | 0.61 | 0.54 | 0.58 | 0.62 | | −900 | 0.71 | 0.58 |
| −0.10 | 0.72 | 0.62 | 0.54 | 0.58 | 0.64 | | −1000 | 0.72 | 0.58 |
| −0.15 | 0.74 | 0.64 | 0.54 | 0.60 | 0.67 | | −1500 | 0.73 | 0.62 |
| −0.20 | 0.78 | 0.64 | 0.49 | 0.59 | 0.70 | | −2000 | 0.72 | 0.64 |
| −0.25 | 0.85 | 0.65 | 0.49 | 0.56 | 0.71 | | −2500 | 0.71 | 0.67 |
| −0.30 | 0.89 | 0.67 | 0.49 | 0.54 | 0.73 | | −3000 | 0.69 | 0.68 |
| −0.35 | 0.86 | 0.68 | 0.49 | 0.52 | 0.79 | | −3500 | 0.66 | 0.67 |
| −0.40 | 0.81 | 0.64 | 0.49 | 0.49 | 0.84 | | −4000 | 0.62 | 0.67 |
| −0.45 | 0.78 | 0.59 | 0.49 | 0.48 | 0.85 | | −4500 | 0.58 | 0.67 |
| −0.50 | 0.72 | 0.57 | 0.49 | 0.48 | 0.83 | | −5000 | 0.54 | 0.66 |

Legend:
- 0.50 - 0.60
- 0.60 - 0.70
- 0.70 - 0.80
- 0.80 - 0.90

**Figure 3.** Overall accuracy of tested indices at different thresholds.

In Figure 4 the Kappa index results are presented for the seven tested indices. The tested thresholds are the same that for overall accuracy. The best Kappa index result (0.77) is for NDWI index with −0.30 threshold. The other indices showed lower Kappa index results (≤0.70) for all the tested thresholds.

| Thresholds | NDWI | MNDWI | CEDEX | RE−NDWI | B_Blue | | Thresholds | Index | |
|---|---|---|---|---|---|---|---|---|---|
| | | | | | | | | Awei (sh) | Awei (nsh) |
| 0.50 | 0.07 | 0.04 | 0.23 | 0.00 | 0.07 | | | | |
| 0.45 | 0.08 | 0.05 | 0.23 | 0.00 | 0.07 | | | | |
| 0.40 | 0.09 | 0.05 | 0.23 | 0.00 | 0.07 | | −50 | 0.24 | 0.05 |
| 0.35 | 0.09 | 0.05 | 0.23 | 0.00 | 0.08 | | −100 | 0.25 | 0.05 |
| 0.30 | 0.10 | 0.05 | 0.23 | 0.00 | 0.08 | | −200 | 0.29 | 0.06 |
| 0.25 | 0.12 | 0.05 | 0.23 | 0.00 | 0.10 | | −300 | 0.32 | 0.06 |
| 0.20 | 0.14 | 0.06 | 0.23 | 0.01 | 0.10 | | −400 | 0.33 | 0.07 |
| 0.15 | 0.18 | 0.10 | 0.23 | 0.02 | 0.11 | | −500 | 0.35 | 0.08 |
| 0.10 | 0.19 | 0.13 | 0.23 | 0.05 | 0.12 | | −600 | 0.36 | 0.09 |
| 0.05 | 0.25 | 0.17 | 0.23 | 0.09 | 0.14 | | −700 | 0.38 | 0.12 |
| 0.00 | 0.31 | 0.19 | 0.23 | 0.12 | 0.19 | | −800 | 0.40 | 0.13 |
| −0.05 | 0.39 | 0.22 | 0.08 | 0.16 | 0.25 | | −900 | 0.43 | 0.15 |
| −0.10 | 0.44 | 0.25 | 0.08 | 0.16 | 0.28 | | −1000 | 0.43 | 0.16 |
| −0.15 | 0.47 | 0.28 | 0.08 | 0.21 | 0.34 | | −1500 | 0.45 | 0.23 |
| −0.20 | 0.56 | 0.28 | −0.02 | 0.17 | 0.39 | | −2000 | 0.45 | 0.29 |
| −0.25 | 0.70 | 0.30 | −0.02 | 0.13 | 0.42 | | −2500 | 0.42 | 0.33 |
| −0.30 | 0.77 | 0.33 | −0.02 | 0.07 | 0.47 | | −3000 | 0.39 | 0.35 |
| −0.35 | 0.72 | 0.35 | −0.02 | 0.04 | 0.58 | | −3500 | 0.33 | 0.35 |
| −0.40 | 0.62 | 0.27 | −0.02 | −0.02 | 0.69 | | −4000 | 0.24 | 0.34 |
| −0.45 | 0.56 | 0.19 | −0.02 | −0.05 | 0.70 | | −4500 | 0.16 | 0.34 |
| −0.50 | 0.43 | 0.13 | −0.02 | −0.04 | 0.66 | | −5000 | 0.09 | 0.32 |

Legend:
- −1.00 - 0.00
- 0.00 - 0.10
- 0.10 - 0.20
- 0.20 - 0.30
- 0.30 - 0.40
- 0.40 - 0.50
- 0.50 - 0.60
- 0.60 - 0.70
- 0.70 - 0.80

**Figure 4.** Kappa index of tested indices at different thresholds.

The performance of the indices can be graphically observed in detail in Figure 5. This figure shows a compilation of images of the indices for their optimal thresholds in a specific area of the outlined water bodies. According to these results, the NDWI index has the best performance for identifying wetland water bodies. One of the factors that could explain the different indices performance can be the lower spatial resolution (20 m) for the bands B5, B11, and B12 used for the indices MNDWI, CEDEX, RE-NDWI, AWEI(SH), and AWEI(NSH). In addition, it was detected that the reflectance values of the bands 11 and 12 on water areas were more variable than B3 and B8 used in the NDWI index.

For further analysis of the NDWI performance, Figures 6 and 7 show detailed overall accuracy and Kappa index results for each wetland and date.

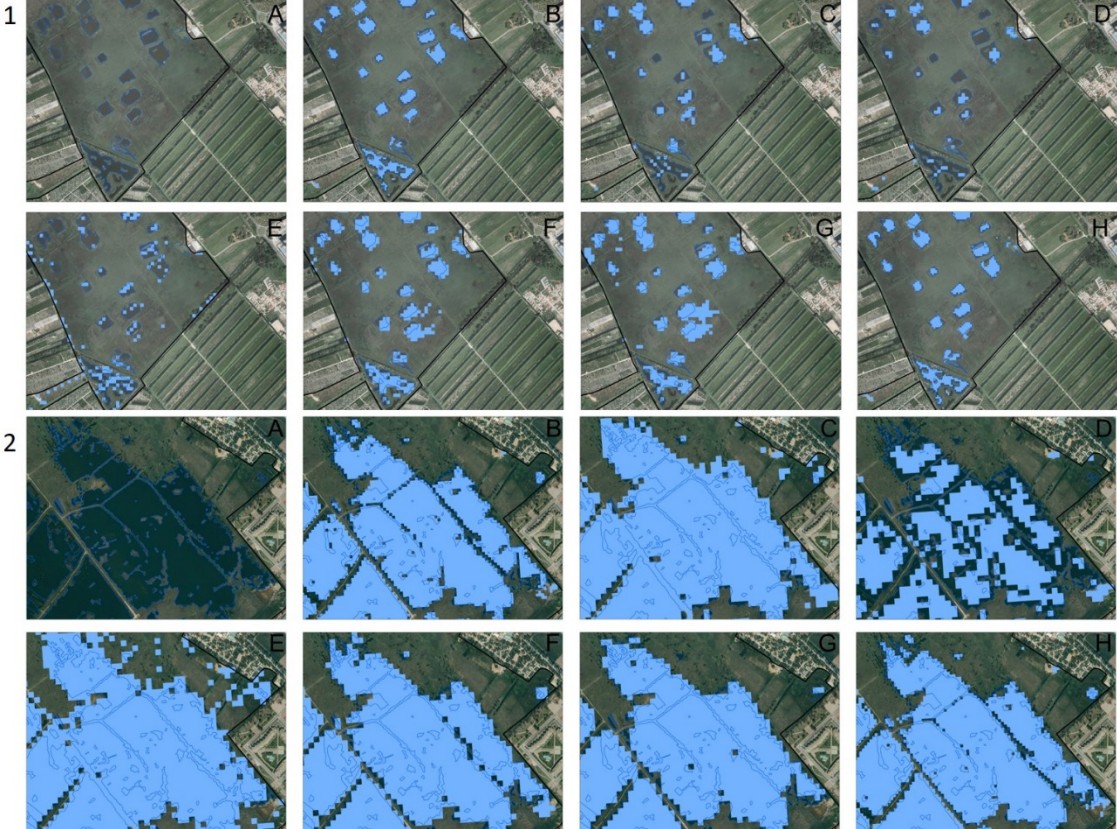

**Figure 5.** Delimited water bodies in two wetlands are represented: (1) Safor wetland 2017 and (2) Prat Cabanes-Torreblanca wetland 2018. (**A**): orthophoto, (**B**): NDWI index (threshold −0.30), (**C**): mNDWI index (threshold −0.35), (**D**): CEDEX index (threshold +0.4), (**E**): Re-NDWI (threshold −0.15), (**F**): Awei sh index (threshold −1500), (**G**): Awei nsh index (threshold −3000) and (**H**): NDWI/BLUE-NIR index (threshold −0.45).

| | | NDWI (Global accuracy) | | | | | | | | |
|---|---|---|---|---|---|---|---|---|---|---|
| | | Cabanes−Torreblanca | | Sagunto | | | La Safor | | | Pego−Oliva | |
| | | 2018 | 2017 | 2018 | 2017 | 2016 | 2018 | 2017 | 2016 | 2018 | 2016 |
| **Thresholds** | 0.50 | 0.50 | 0.50 | 0.50 | 0.50 | 0.52 | 0.50 | 0.77 | 0.57 | 0.50 | 0.52 |
| | 0.45 | 0.50 | 0.50 | 0.50 | 0.50 | 0.52 | 0.50 | 0.78 | 0.58 | 0.50 | 0.52 |
| | 0.40 | 0.50 | 0.50 | 0.50 | 0.50 | 0.52 | 0.50 | 0.80 | 0.60 | 0.50 | 0.52 |
| | 0.35 | 0.50 | 0.50 | 0.50 | 0.50 | 0.52 | 0.50 | 0.80 | 0.60 | 0.50 | 0.55 |
| | 0.30 | 0.50 | 0.50 | 0.50 | 0.50 | 0.52 | 0.52 | 0.80 | 0.63 | 0.50 | 0.55 |
| | 0.25 | 0.50 | 0.50 | 0.50 | 0.50 | 0.52 | 0.53 | 0.80 | 0.70 | 0.50 | 0.55 |
| | 0.20 | 0.52 | 0.50 | 0.50 | 0.50 | 0.52 | 0.55 | 0.80 | 0.75 | 0.50 | 0.55 |
| | 0.15 | 0.60 | 0.53 | 0.50 | 0.50 | 0.52 | 0.60 | 0.80 | 0.77 | 0.50 | 0.57 |
| | 0.10 | 0.63 | 0.55 | 0.50 | 0.50 | 0.52 | 0.60 | 0.82 | 0.77 | 0.50 | 0.57 |
| | 0.05 | 0.70 | 0.70 | 0.50 | 0.50 | 0.52 | 0.60 | 0.83 | 0.78 | 0.52 | 0.62 |
| | 0.00 | 0.72 | 0.77 | 0.50 | 0.50 | 0.55 | 0.72 | 0.85 | 0.80 | 0.52 | 0.63 |
| | −0.05 | 0.80 | 0.82 | 0.53 | 0.55 | 0.55 | 0.77 | 0.93 | 0.80 | 0.52 | 0.67 |
| | −0.10 | 0.85 | 0.88 | 0.55 | 0.55 | 0.55 | 0.80 | 0.93 | 0.80 | 0.53 | 0.73 |
| | −0.15 | 0.83 | 0.90 | 0.55 | 0.63 | 0.57 | 0.85 | 0.93 | 0.83 | 0.53 | 0.73 |
| | −0.20 | 0.85 | 0.93 | 0.62 | 0.68 | 0.60 | 0.90 | 0.95 | 0.83 | 0.63 | 0.78 |
| | −0.25 | 0.88 | 0.97 | 0.73 | 0.75 | 0.78 | 0.97 | 0.97 | 0.85 | 0.83 | 0.78 |
| | −0.30 | 0.93 | 0.95 | 0.80 | 0.83 | 0.87 | 0.97 | 0.97 | 0.88 | 0.90 | 0.75 |
| | −0.35 | 0.88 | 0.90 | 0.78 | 0.88 | 0.90 | 0.93 | 0.93 | 0.87 | 0.82 | 0.72 |
| | −0.40 | 0.73 | 0.85 | 0.72 | 0.90 | 0.78 | 0.93 | 0.92 | 0.85 | 0.73 | 0.70 |
| | −0.45 | 0.62 | 0.80 | 0.75 | 0.87 | 0.77 | 0.90 | 0.88 | 0.83 | 0.72 | 0.65 |
| | −0.50 | 0.52 | 0.67 | 0.70 | 0.75 | 0.68 | 0.87 | 0.83 | 0.85 | 0.63 | 0.67 |

0.50 - 0.60
0.60 - 0.70
0.70 - 0.80
0.80 - 0.90
0.90 - 1.00

**Figure 6.** Overall accuracy of NDWI index tested at four studies areas and 3 years for the threshold range (−0.50 to 0.50).

| NDWI (Kappa index) | | | | | | | | | | |
|---|---|---|---|---|---|---|---|---|---|---|
| | Cabanes–Torreblanca | | Sagunto | | | La Safor | | | Pego–Oliva | |
| Thresholds | 2018 | 2017 | 2018 | 2017 | 2016 | 2018 | 2017 | 2016 | 2018 | 2016 |
| 0.50 | 0.00 | 0.00 | 0.00 | 0.00 | 0.03 | 0.00 | 0.53 | 0.13 | 0.00 | 0.03 |
| 0.45 | 0.00 | 0.00 | 0.00 | 0.00 | 0.03 | 0.00 | 0.57 | 0.17 | 0.00 | 0.03 |
| 0.40 | 0.00 | 0.00 | 0.00 | 0.00 | 0.03 | 0.00 | 0.60 | 0.20 | 0.00 | 0.03 |
| 0.35 | 0.00 | 0.00 | 0.00 | 0.00 | 0.03 | 0.00 | 0.60 | 0.20 | 0.00 | 0.10 |
| 0.30 | 0.00 | 0.00 | 0.00 | 0.00 | 0.03 | 0.03 | 0.60 | 0.27 | 0.00 | 0.10 |
| 0.25 | 0.00 | 0.00 | 0.00 | 0.00 | 0.03 | 0.07 | 0.60 | 0.40 | 0.00 | 0.10 |
| 0.20 | 0.03 | 0.00 | 0.00 | 0.00 | 0.03 | 0.10 | 0.60 | 0.50 | 0.00 | 0.10 |
| 0.15 | 0.20 | 0.07 | 0.00 | 0.00 | 0.03 | 0.20 | 0.60 | 0.53 | 0.00 | 0.13 |
| 0.10 | 0.27 | 0.10 | 0.00 | 0.00 | 0.03 | 0.20 | 0.63 | 0.53 | 0.00 | 0.13 |
| 0.05 | 0.40 | 0.40 | 0.00 | 0.00 | 0.03 | 0.20 | 0.67 | 0.57 | 0.03 | 0.23 |
| 0.00 | 0.43 | 0.53 | 0.00 | 0.00 | 0.10 | 0.43 | 0.70 | 0.60 | 0.03 | 0.27 |
| −0.05 | 0.60 | 0.63 | 0.07 | 0.10 | 0.10 | 0.53 | 0.87 | 0.60 | 0.03 | 0.33 |
| −0.10 | 0.70 | 0.77 | 0.10 | 0.10 | 0.10 | 0.60 | 0.87 | 0.60 | 0.07 | 0.47 |
| −0.15 | 0.67 | 0.80 | 0.10 | 0.27 | 0.13 | 0.70 | 0.87 | 0.67 | 0.07 | 0.47 |
| −0.20 | 0.70 | 0.87 | 0.23 | 0.37 | 0.20 | 0.80 | 0.90 | 0.67 | 0.27 | 0.57 |
| −0.25 | 0.77 | 0.93 | 0.47 | 0.50 | 0.57 | 0.93 | 0.93 | 0.70 | 0.67 | 0.57 |
| −0.30 | 0.87 | 0.90 | 0.60 | 0.67 | 0.73 | 0.93 | 0.93 | 0.77 | 0.80 | 0.50 |
| −0.35 | 0.77 | 0.80 | 0.57 | 0.77 | 0.80 | 0.87 | 0.87 | 0.73 | 0.63 | 0.43 |
| −0.40 | 0.47 | 0.70 | 0.43 | 0.80 | 0.57 | 0.87 | 0.83 | 0.70 | 0.47 | 0.40 |
| −0.45 | 0.23 | 0.60 | 0.50 | 0.73 | 0.53 | 0.80 | 0.77 | 0.67 | 0.43 | 0.30 |
| −0.50 | 0.03 | 0.33 | 0.40 | 0.50 | 0.37 | 0.73 | 0.67 | 0.70 | 0.27 | 0.33 |

Legend:
- 0.00 - 0.10
- 0.10 - 0.20
- 0.20 - 0.30
- 0.30 - 0.40
- 0.40 - 0.50
- 0.50 - 0.60
- 0.60 - 0.70
- 0.70 - 0.80
- 0.80 - 0.90
- 0.90 - 1.00

**Figure 7.** Kappa index of NDWI index tested at four studies areas and 3 years for the threshold range (−0.50 to 0.50).

Overall accuracy results are optimal for −0.30 threshold, with 0.89 average value and maximum values of 0.97 in Safor wetland (2017/2018) and Prat Cabanes-Torreblanca (0.93 in 2018 and 0.95 in 2018) (Figure 6). Kappa index results show the highest average value (0.77) for −0.30 threshold. Kappa index results are even better in Cabanes-Torreblanca and Safor wetlands. However, in Sagunto and Pego-Oliva wetland the average Kappa index is lower (0.67 and 0.65 respectively). Considering that the objective is to provide a methodology to characterize water bodies in wetlands as global as possible, this analysis supported the choice of the NDWI index and −0.30 threshold. These results have been obtained for the areas with highest variability, that is the water bodies borders, and could be even more accurate in central areas of land/water covers with less variation in reflectance values.

In Figure 8 we compare the delimited wetland bodies of water in blue color (A, C, E, G) versus the result of the NDWI index for the threshold −0.30 (B, D, F, H) in the four studied wetlands. Since water sampling points are marked in yellow and non-water in red, we can observe false negatives and false positives results. A good performance of the NDWI and −0.30 threshold can be appreciated in pixels that are further from the border (more homogenous areas). In contrast, false positive and false negative pixels were found in the border areas. In the randomly selection of ground truth points within these areas (20 m to the border to each side), some of them were selected randomly at a distance lower than 10 m of the border, and the pixel size of Sentinel-2A/B bands used in NDWI index was 10 m. The classification result of those pixels depends on the area percentage that corresponds to each class. For example, a water ground truth point can be located in a non-water (land) class pixel since the percentage of non-water class for that pixel is higher than for water area (see most of the false negative and false positive in Figure 8).

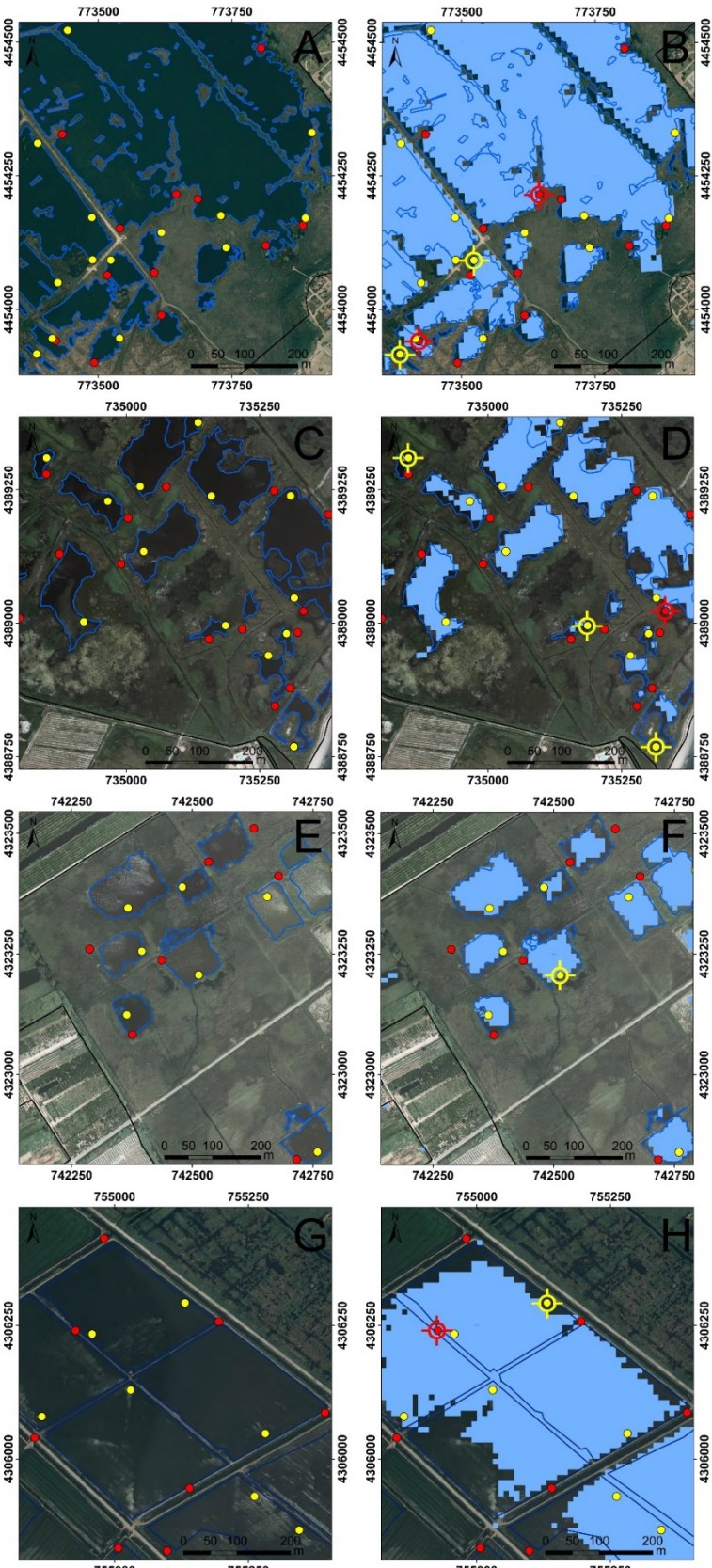

**Figure 8.** Delimited wetland bodies of water in blue color outline (**A,C,E,G**) versus the result of the NDWI index for the threshold −0.30 in light blue (**B,D,F,H**). Water sampling points in yellow and non-water in red. False negatives and false positives are marked. (**A–B**): Prat Cabanes-Torreblanca wetland (2018), (**C–D**): Sagunto wetland (2017), (**E–F**): La Safor wetland (2017) and (**G–H**): Pego-Oliva wetland (2018).

## 4. Discussion

Sentinel-2A/B images spatial resolution allowed to detect smaller water bodies than previously published works [3,6]. The minimum water body surface detected was 100 m². In Figure 9, we show the water bodies identified by Sentinel-2A/B in Safor wetland for the three studied dates. The highest flooded surface was 41.93 ha in August 2017 and the lowest 16.57 ha in November 2016. Precipitation is one of the main variables that determine the extent of the flooded area in this type of wetlands, and 2017 was the rainiest period studied (Table 1). The flooded surface is not necessarily a continuum but the addition of water bodies that can be connected or not. The average size of flooded areas is around 1500 m², so the use of Sentinel-2A/B images is key for detecting these water bodies and avoid underestimating the water surface. It is important to highlight that the NDWI index is not able to detect the water layer underlying marsh vegetation (e.g., *Phragmites australis*) but free water layers. The Ramsar Convention established a Wetland Type Classification System that identifies 42 types grouped into three main categories: marine and coastal, continental, and artificial. Thus, a certain wetland area is identified with a main type of wetland according to its predominance, but it may have more than one type present, this being the most common. The selected study areas are coastal wetlands, and in Table 2 we summarize the habitats present. The main free water layers are coastal lagoons, natural eutrophic lakes, and natural dystrophic lakes and ponds. Figure 9 shows the habitats present in Safor wetland, water areas are characterized by its size, morphology, vegetation, and period of permanence of the water layer. For instance, natural lakes (locally known as "ullals") have a permanent water surface along the year, but grasslands are vegetation only periodically inundated. This great variability makes it difficult to monitor the status of these wetlands, mainly due to the small size of some surfaces that makes detection and quantification difficult [37]. Following the described methodology, we have been able to detect all these water surfaces under different inundation conditions.

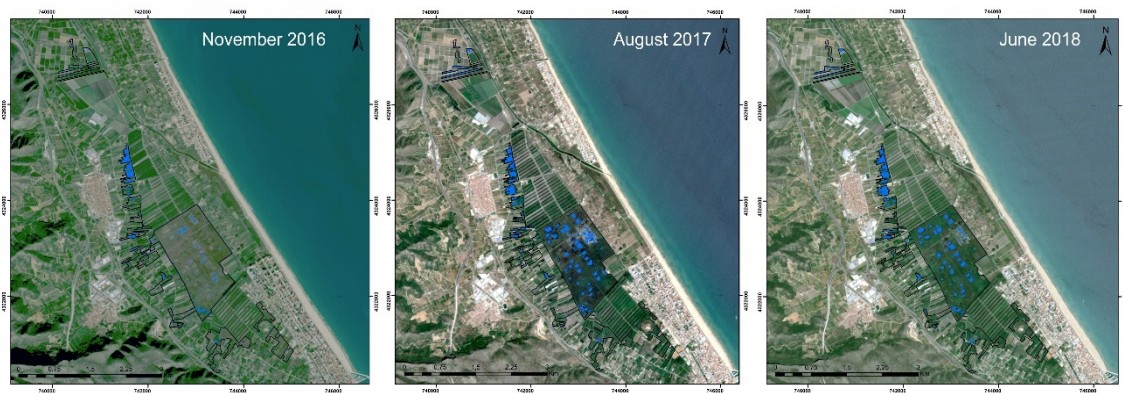

**Figure 9.** Delimited water bodies with the NDWI index in Safor wetland.

In our study the best performing index was the NDWI index [31]. This result is in line with previous studies indicating the high performance of this index for extracting water surfaces [11,38–40]. The results obtained in our study are significant in at least two major aspects. The first one is related to the proposal of a stable and generalizable threshold to delineate water surfaces in the studied coastal wetlands. These results differ from earlier findings indicating that the threshold values for classifying water from non-water are unstable and vary according to date, location, and the subpixel land-cover components [17,41,42]. Other authors applied other indices such as Awei(sh) and MNDWI to improve NDWI performance. However, this improvement is only observed in areas including dark built surfaces and buildings, because the NDWI often does not distinguish between water areas and built-up land [17,32]. It is important to remark that before applying the NDWI index it is necessary to delimitate the natural wetland area. These ecosystems usually hold protection figures and have an official cartography that can be used as delimitation. Some studies reported that limited spatial resolution of some satellite images (Landsat TM, 30 m, MODIS, 250 m, AVHRR 1 km, SPOT vegetation 1 km) can contain a mixture of land-cover types being this effect more significant in

edge pixels [41] and generating difficulties for mapping homogeneous coastal wetlands [43,44]. Ji et al. [41] demonstrated that the threshold for extracting water surfaces depends on the subpixel land-cover components. Some authors suggest the categorization of surface water for image classification [42,44]. In this study, the higher spatial resolution of the Sentinel-2A/B bands (10 m for band 3, Green, and band 8, NIR) allowed to define with more accuracy the boundary of water surfaces avoiding pixels with a mixture of covers (soil, vegetation, water). The minimum surface water areas detected on this study could not be extracted using moderate spatial resolution images such as Landsat [45]. Other studies have also observed a better performance with Sentinel-2A/B images than Landsat images [46].

The second aspect of this study that should be highlighted is the threshold sign. Early research on this topic proposed a threshold of 0 for the water indices NDWI and MNDWI [31,32]. Values greater than 0 were classified as water pixels and values lower than 0 as non-water pixels [31,32]. However, what stands out in our analysis was that the best performing thresholds were negative values. This finding is consistent with other studies where values lower than 0 were set for extracting water bodies [38–41]. Ji et al. [41] reported that the mixture of land covers distributed in the same pixel had a strong impact on the NDWI values obtaining negative and variable thresholds according to the relative proportions of soil and vegetation. The negative values for extracting water surfaces may be explained considering the spectral response of the analyzed wetlands. Water quality has a significant effect on reflectance, high values of chlorophyll *a* generate higher reflectance values for the NIR band (B3) than for the GREEN band (B8) (Table 2). Consequently, the negative values of the selected threshold are congruous with the spectral response of the water in the analyzed wetlands. In the studied areas, eutrophic and dystrophic lakes water can be classified as complex waters with variable concentration of chlorophyll *a* and other colored substances such as humic acids.

The majority of studies developed in wetlands have focused on calculating global gain or loss of wetland area and have found that agricultural and urban land use conversion are the main causes of wetland loss [47]. This is critical to the success of the no-net-loss wetland conservation international strategy. Additionally, inside of wetlands which hold protection figures, land use change in the surrounding area has a direct effect on water management strategies. In the studied areas, especially in Safor wetland, agricultural and urban use coexist with natural use, and that strongly condition water management. Nowadays, Safor wetland hydrology is anthropogenically manipulated to prevent crop root asphyxia, and to avoid flooding of urban areas. In the wet seasons, water is pumped through irrigation channels into the sea to decrease the phreatic level [24]. This manipulation can produce important impacts in the hydrologic cycle of the wetland that have not been studied. Applying Sentinel-2A/B images to monitor flooded area changes trough the NDWI index will be key to analyze the consequence of these management actions. This data can be used to assist both wetland managers and practitioners to make decisions about priority management interventions to maintain the ecological character of a wetland. As other studies have already pointed out [5], the use of Earth Observation tools is key for addressing the information gaps faced by wetland managers and practitioners.

## 5. Conclusion

The results of this study indicated the potential of NDWI index calculated from Sentinel-2A/B images (bands 3 and 8) to extract open water bodies in delimited wetlands. It was observed that a −0.30-threshold generated acceptable results to classify the studied coastal wetlands. This threshold results are proper along the year under different flooding and vegetation conditions, allowing to distinguish water and non-water (soil, vegetation) polygons. The spatial resolution of these images allowed to detect water bodies of reduced size (the average size of flooded areas is around 1500 m$^2$) compared to previous missions of medium and low resolution. In the studied wetlands, the flooded surface is not necessarily a continuum but the addition of water bodies that can be connected or not depending partly on the pluviometry regime. The information derived from Sentinel-2A/B bands can be very useful to monitor these ecosystems, offering valuable information for managers of these areas, specially to study the effect of hydrologic cycle manipulation.

**Author Contributions:** Conceptualization, J.P.R., M.T.S.F., J.E. and J.A.A.M.; Data curation, J.P.R., M.T.S.F., J.E. and J.A.A.M.; Formal analysis, J.P.R.; Investigation, J.P.R., M.T.S.F., J.E. and J.A.A.M.; Methodology, J.P.R., M.T.S.F., J.E. and J.A.A.M.; Software, J.P.R.; Supervision, M.T.S.F. and J.E.; Validation, M.T.S.F. and J.E.; Visualization, M.T.S.F. and J.E.; Writing – original draft, J.P.R., M.T.S.F., J.E. and J.A.A.M.; Writing – review & editing, M.T.S.F. and J.E.

**Funding:** The authors declare no funding has been received to support this research.

**Conflicts of Interest:** The authors declare no conflict of interest.

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
