# Peer review of "Sentinel-2 Application to the Surface Characterization of Small Water Bodies in Wetlands"

_water, doi:10.3390/w12051487_

Round 1
Reviewer 1 Report
Review of the manuscript entitled “Sentinel-2 application to the surface characterization of small water bodies in wetlands” by Jesús Pena-Regueiro et al.
This study consists on 1) testing Sentinel-2A/B images suitability for detecting small water bodies in wetlands characterized by high diversity of temporal and spatial flooding patterns using previously published indexes; 2) application on medium spatial resolution Sentinel-2A/B images of four coastal wetlands in the Valencia Region (East Spain, Mediterranean Sea); and 3) Validation of the small water body detection using images of high spatial resolution from the Valencian Cartography Institute (ICV) Orthophoto.
Authors state that the best performing index was the NDWI. Thus they propose a new stable and generalizable threshold to delineate water surfaces in coastal wetlands.
No doubt, this is a relevant topic for the hydrology and remote-sensing community; particularly wetland managers and policy makers for flood monitoring.
Although this subject is large, authors have succeeded to synthesis the literature and provide a fair overview of previous studies.
Authors set an effort to explain their method, which I believe that, they make it clear to the reader and easy to be followed starting by the description of the studied areas, then by the data processing.
Clearly, authors deticated lots of work to get these results, which are valuable for EO communities.
The main concerns of this paper are:
- The manuscript misses the conclusion to summarize the main finding of this study.
- Readers miss clear interpretation of the results (Figure 3-6) w.r.t the index computing and values
- Examples of the flood impact on the natural habitats and the ecosystem of the studied area.
Although these concerns (above) and few typo corrections and comments (below), I would recommend this manuscript for publication with minor revision.
Minor comments:
Note that P: stands for page; L: stands for line.
P4,L20: why the authors used Sentinel-2 data from two different sources?
P6, Table-4 , In MNDWI, used B11 for SWIR2, should be B12.
P6,L152: Authors need to justify the choice of 60 & 30 points for the validation.
P6,L159: Accuracy used several times in the manuscript, but has not been defined anywhere. It is worth to define/clarify which type of accuracy been used and add reference if possible.
P6,L161: Knowing that Kappa does not quantify the level of agreement between two datasets, It represents the level of agreement of two dataset; and since there is different Kappa’s, Authors need to provide a reference of the equation-1, (e.g. See Olofsson, Foody, Herold, Stehman, Woodcock and Wulder (2014, Remote sensing of Environment)).
P6,L177: mismatching between index and Figure 3 & 4 (B-Blue).
P10,L247-252: I think these lines should be moved to the introduction.
Note: I have not time to go through the references, I trust the authors and the Ed-team for a re-read/review them
Author Response
Dear reviewer,
thank you very much for your detailed review.
Please, find attached the response letter.
yours,
Maite

Reviewer 2 Report
The manuscript ”Sentinel-2 application to the surface characterization of small water bodies in wetlands” evaluates the use of Sentinel-2 data to delineate small areas of standing water ~ 100 m2 in wetland regions. They analyzed the use of several common literature indices that have previously been used to categorize water. They used high-resolution aerial images as ground-truth. They round the NDWI with a negative threshold of -30 provided the most accurate method.
I found the manuscript useful, though I doubt the results can be generalized, but rather are specific to similar wetlands.
A general comment; the data clearly show that NDWI provides the best indicator of water for the three wetlands studied. However, there is no discussion of false positives or of other types of wetlands, especially in areas outside of the wetlands. Wetlands generally have two types of land cover, either heavy vegetation or water. In most wetlands, the vegetation is essentially uniform, usually a variety of phragmites or other rushes. In some wetlands, the vegetation is grasses. While in others, the vegetation may be a mix of grass and rushes. Depending on the area, there may also be bare soil. I think the study is useful and should be published, but the authors should be careful about making claims that the value of -30 is a good “general” value. Other studies of NDWI, including those reported by this paper, used a much higher index value to delineate water. I suggest the authors reconsider some of the “general applicability” statements and make more narrow statements.
The paper needs some “high resolution” images where we can see both the ground truth and the results of the classification algorithm. It would be useful to show both successful, false positive, and false negative pixels. I would also like to see a high-resolution example of the ground truth points locations.
Since the authors stated they used a GIS program to delineate water areas in the images, why were not all the pixels categories as either water or non-water and used in the accuracy assessment. The manuscript notes that only a random selection of points were used.
I provided some markups that include indicating a few typos in the manuscript.
Specific comments
L15 The abstract does not discuss ground truth – should be added.
L130 Need resolution of ground truth images (Table 3)
L134 Does not discuss how water and non-water polygons were delineated. I assume through visual examination, but it should be stated. A high-resolution area of one of the photos providing an example would help here.
L136 Stated “high resolution” but the actual resolution is not provided.
L162 Equation. The description of the variables in not correct, needs to be fixed.
L180 – 200. Figures 3 – 6. This is very interesting data. I would like more discussion on how the various indexes got things wrong, along with some high-resolutions (pixel level) images showing these issues. For example, were the poorly performing algorithms missing with false positives or false negatives or were the errors of both types. Were there any general conclusions that could be drawn – why for example is the RE-NDWI so bad for wet-lands, when it performs well in other situations. What is unique to this environment that makes it bad.
L238 – 240 You indicate that previous results showed changes in the index threshold value through time. I would expect some of this would be due to changing vegetation colors (e.g., high chlorophyll (green) to low chlorophyll (brown to yellow). Your data were essentially taken in June and July, with the exception of on image in November. Was the November image classified as accurately as the summer images? Also, I’m not familiar with the region, but do the wetland plants turn brown in the winter? In my area they do, but in warmer areas they are green throughout the year. This is an example of something that might not generalize and show why your threshold value did not change through the year, while other studies showed a change.

Author Response
Thank you very much for your detailed revision. Please, find in the attached file our revision answers.

Reviewer 3 Report
In general, the paper is robust and the only things that I recommend are a) to enrich the introduction based on relevant studies that have the same target and to give some information about the current state-of-the-art, b) to create a seperate conclusion section in order to follow the format specifications of the journal.
By reading the title of the paper, I would expect to see a methodology for identidying small water bodies based on larger pixel resolutions of EO data using resolution enhancement methods. But reading the abstract and the paper, I understood that the paper is just an exercise of using old indices for identifying water bodies >100 m2, which are inside the resolution capacity of Sentinel 2. So why exactly is stressed the word "small" in the title and what is new in the methodology of this paper when new methods of resolution enhancement is now the state-of-the-art? I would expect some discussion on this issue. Resolution enhancement is something that can be done by everyone? what sources are needed? there were such sources available in this study in order to do such analyses?
Author Response

(The authors gave the same response as above.)

Round 2
Reviewer 3 Report
The authors addressed adequately the comments. The manuscript can be accepted.
Author Response
Dear reviewer,
thank you very much for appreciating our effort
Yours,
Maite